# How Do Ecological Restoration Projects Affect Trade-Offs and Synergies between Ecosystem Services?

Yuhui Ji [1], Miaomiao Xie [1,2,*], Yunxuan Liu [1], Renfen Zhu [1], Zhuoyun Tang [1] and Rongwei Hu [1]

1   School of Land Science and Technology, China University of Geosciences, Beijing 100083, China; 2112210052@email.cugb.edu.cn (Y.J.); 3012230014@email.cugb.edu.cn (Y.L.); 2012230046@email.cugb.edu.cn (R.Z.); 2012200045@email.cugb.edu.cn (Z.T.); 2112220051@email.cugb.edu.cn (R.H.)
2   Key Laboratory of Land Consolidation, Ministry of Natural Resources of the PR China, Guanying Yuan West 37, Beijing 100035, China
*   Correspondence: xiemiaomiao@cugb.edu.cn; Tel.: +86-010-8232-1807

**Abstract:** Scientific ecosystem management requires the clarification of the synergic and trade-off relationship between ecosystem services, particularly in the environmentally delicate Loess Plateau region. Previous studies have indirectly deduced that ecological restoration projects affect ESRs by analyzing their impacts on ecosystem services, but there is no direct evidence from the existing research to show whether and to what extent different ecological restoration projects have an impact on trade-off synergies, which weakens the explanatory strength of ecological restoration projects as an important factor affecting ESRs. In this study, based on the spatial mapping of three pairs of relationships between three typical ESs in Fugu County, Shaanxi Province, and the relative contribution of each ecological restoration projects, as well as Ecosystem services and the relationship between them, were explored through the boosted regression tree modeling (BRT). This study proved that different ecological restoration projects have different impacts on ESRs. The results indicated that the three pairs of ESRs obtained among the three ecosystem services in Fugu County could be categorized into two types. The relationship between carbon storage and soil conservation and the relationship between carbon storage and water conservation CS–WC were spatially predominantly trade-offs, and their spatial distributions were highly similar. Various ecological restoration projects have varying effects on ESRs. The connection between ecological restoration projects and ESRs involves a nonlinear transformation, and the change varies from project to project. Based on the above findings, this study further explores the influence process of various types of ecological restoration projects on ESRs, and provides scientific support for optimizing ecosystem management and comprehensive management of the region.

**Keywords:** ecosystem services; trade-offs and synergies; ecological restoration; Loess Plateau





## 1. Introduction

The "2021–2030 United Nations Decade of Ecosystem Restoration" seeks to stop, prevent, and reverse environmental harm. Ecological restoration promotes the restoration and maintenance of biodiversity, thereby increasing ecosystem stability and sustainability, and providing more ecosystem services to humans. Ecosystem services (ESs) are the advantages that ecosystems offer to human well-being [1,2]. Under the influence of other factors, such as climatic conditions and human activities, the relationships between multiple ecosystem services are characterized by both trade-offs and synergies. Ecological restoration works restore original ecological functions and biodiversity by rehabilitating and reconstructing ecosystems that have been degraded by human activities. The goal of such work is to maintain the health and sustainability of ecosystems while conserving biodiversity and providing ecosystem services needed by humans. Over the past few decades, ecosystems around the globe have been severely damaged and degraded due to human activities.

As a result, research on ecological restoration engineering and ecosystem services has received increasing attention from scientists and policymakers. At present, with the rapid development of industry and agriculture, the degradation of ecosystems is increasing, and people are gradually recognizing the importance of ecological restoration. China, as one of the countries that adopted ecological restoration earlier, has benefited from implementing several ecological conservation and restoration projects since the 1950s. Evaluating the changes in ecosystem service relationships (ESRs) and investigating the effects of various ecological restoration projects on them can help optimize ecosystem management and establish a scientific basis for the region's overall management, especially in light of the current need for ecological protection and sustainable development.

Most scientists and policymakers agree that trade-offs and synergies among ecosystem services are important [3]. Different ESs are interrelated rather than independent, and their nonlinear relationships are categorized into ESRs that gain from each other [2,4,5]. Over the past several decades, most of the research has focused on quantitatively assessing ecosystem services and identifying the trade-offs and synergies among various services [6]. Relationships between ecosystem services are not invariant, and changes between them can vary with temporal and spatial scales [2,7–9]; furthermore, neglecting the spatial and temporal dimensions may result in a potential misunderstanding of the co-occurrence of ecosystem services in both space and time [2,10]. Therefore, it is more comprehensive and reliable to study ESRs from both temporal and spatial variation perspectives at the same time.

Ecological restoration projects cause a redistribution of soil and water resources by altering surface hydrological processes, which can, in turn, reduce the negative impacts on the ecosystem during social development [11,12]. ESRs are also influenced by socioecological factors, including ecological restoration projects [13]. Different ecological restoration measures may have different impacts on ecosystem services as well as ESRs. For example, in Zhang et al.'s study, ecological restoration projects were shown to improve greening and stewardship services, which have a positive impact on enhancing ecosystem services; however, it was underlined that greening exacerbates trade-offs between socioecological systems in degraded ecoregions [14]. Pan et al.'s study, assessing and comparing the impacts of tree plantations and primary forests, demonstrated that ecological restoration projects may not only enhance ecosystem services, but may also bring about some environmental and social trade-offs [15]. Forest restoration policies such as China's program to return farmland to forest may increase afforestation and increase carbon sequestration. At the same time, the replacement of farmland may lead to a decrease in food production, thus creating a trade-off between two ecosystem services: carbon sequestration and food production [16]. Ecological restoration projects can help to enhance the overall ecosystem services, but the complex trade-offs and synergistic relationships between different services may affect the effectiveness of the projects. Failure to consider these complex effects of ecological restoration projects may lead to unwise management decisions that are not conducive to the enhancement of ecosystem services. The current research gap in studies addressing ecological restoration projects as a driver to explore ES trade-off synergies is that there is no direct evidence on whether and to what extent different ecological projects have an impact on trade-off synergies. There are no studies that directly use ESRs as a dependent variable [5,17], which weakens the explanatory strength of ecological engineering as an important factor influencing trade-off synergies. There is no direct evidence in the current research on whether and to what extent different ecological engineering projects have an impact on trade-off synergies.

Another research deficiency is about the research methodology. Correlation analysis is now employed to investigate the intricate trade-offs and synergistic interactions between ESs [18–21], Bayesian belief networks [5,22], or spatial mapping [23], among others. Correlation analysis can only determine the global synergy of trade-offs, while the spatial mapping approach is unable to identify the features of local variations as well as the spatial locations of trade-offs and synergies, which are difficult to couple with other elements for

subsequent analysis. By comparing the positive and negative changes in two different ESs in different years, the difference comparison approach discerns whether the link between ESs is synergistic or a trade-off [6]. ESRs obtained using this method can reflect spatial heterogeneity, while the result is a binary variable that can be analyzed by coupling with other elements, which facilitates the subsequent coupling of the trade-off synergistic relationship with ecological restoration projects to analyze its impact mechanism. Most of the current studies that have used ESRs as the object of study to examine specific impact processes have used linear regression models such as logistic regression [23] and geographically weighted logistic regression [6]. However, ecosystems are characterized by multistability and nonlinear changes [5,24], and the impacts of ecological restoration projects on ecosystems also change nonlinearly, so nonlinear models are more in line with the actual situation than linear models.

The Loess Plateau region has long been characterized by ecological problems such as low vegetation cover, scarcity of water resources, and serious soil erosion. Because of this, it is a crucial area for carrying out ecological restoration projects. Extensive ecological restoration projects have notably enhanced the ecological conditions of the Loess Plateau, leading to substantial enhancements in various ecosystem services, including water and soil conservation, as well as carbon sequestration [25,26]. With the increasing number of ecological restoration projects implemented in the Loess Plateau region, researchers discovered that the Loess Plateau Vegetation Conservation Project further improved the synergistic interaction between different ecosystem services by altering the number of ESs [27]. Nevertheless, carrying out ecological restoration projects may also result in issues such as decreased surface runoff, potentially causing water conflicts. Fu et al. argue that the implementation of the Grain for Green projects will result in trade-offs between the provision of carbon sequestration and water production services, and that regional water resources on the Loess Plateau are about to be unable to carry the current scale of implementation of vegetation restoration projects [25,28]. It has also been pointed out that some of the vegetation restoration projects will also use an increased amount of water, thus resulting in the issue of ecosystem service trade-offs [24,29]. Therefore, understanding the synergistic effects of trade-offs arising from the implementation of these ecological restoration projects in the Loess Plateau region is important for optimizing the implementation of ecological restoration projects and maximizing the comprehensive benefits of ecosystem services.

Therefore, this paper uses an augmented regression tree model to explore how ecological restoration projects affect ESRs, which is a powerful machine learning model that can better explain and predict the interrelationships between variables that are nonlinear and predict nonlinear changes. in a Loess Plateau county. This study has three main goals: to (1) explore the features of the ESRs' temporal and spatial divergence among Fugu County's for three ecological services from 1990 to 2020, (2) analyze the relative impacts of various ecological restoration projects on ESRs, and (3) to describe whether the effects of different ecological restoration projects on ESRs are nonlinear or linear, as well as how ecological restoration projects affect the changes in ESRs. We believe that the main innovation of this paper is the use of a nonlinear coupled model, which directly proves that ecological restoration projects will have different nonlinear impacts on ESRs and can provide a reference for the scientific formulation of ecological restoration policies.

## 2. Materials and Methods

### 2.1. Study Area

Fugu County is situated in the northern region of Yulin City, Shaanxi Province, with an area of about 3201 km$^2$, and is positioned at the border between the Inner Mongolian Plateau and the northeastern Loess Plateau in the northern part of Shaanxi (Figure 1). The terrain is high in the northwest and low in the southeast, mainly by the northwest-to-southeast flow of the Huangfu River, Qingshui River, Gushan River, Shima River, and Beiniu River and the corresponding five mounts composing the main geomorphological

skeleton, with an elevation of 780.0~1426.6 m. The five major rivers of Fugu County belong to a mesothermal semiarid continental monsoon climate, with large temperature variations and large interannual variations in precipitation. The average yearly temperature is around 9 °C, and the average annual rainfall is approximately 400 mm. Fugu County is a major coal-producing county in China, and the problem of mining subsidence caused by continuous coal mining has seriously affected people's productivity and lives. Fugu County has long-term issues with soil erosion and is located in the middle reaches of the Yellow River, which is a concentrated source area of coarse silt. Fugu County has been a significant site for ecological restoration projects, with the construction of numerous check dams and extensive ecological restoration projects including afforestation, grass planting, and land sealing since the 1970s. Additionally, various other projects for ecological restoration and management, such as slope cultivation and land remediation, have been undertaken in the area. Fugu County can be divided into five large basins according to the location of five rivers.

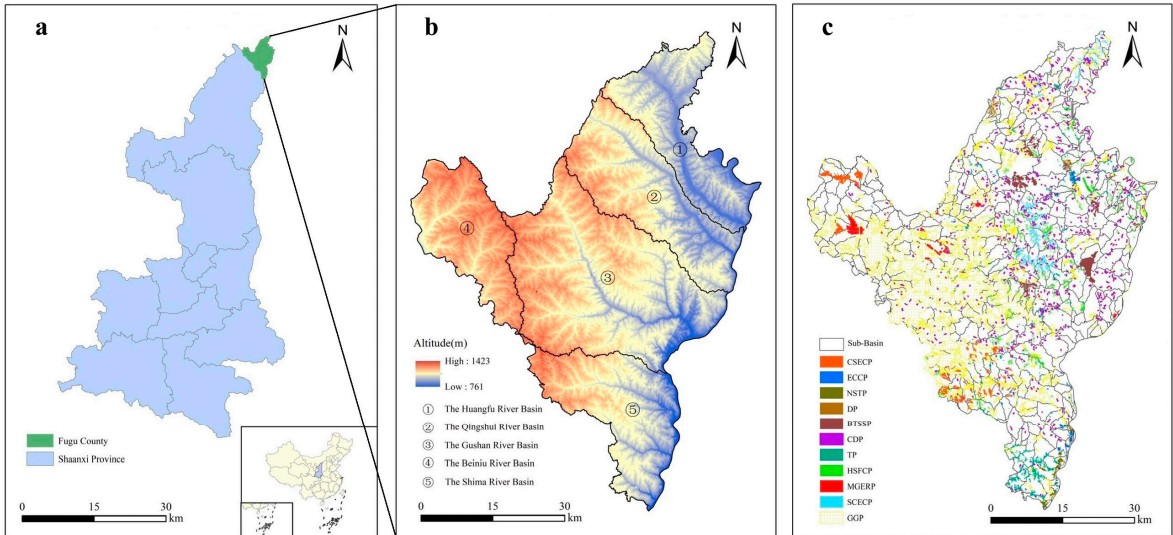

**Figure 1.** Overview of the study area. (**a**) Location of the study area; (**b**) altitude; (**c**) ecological restoration projects. GGP: the Grain for Green Project; CDP: Check Dam Project; SCECP: Sloping Cropland Ecological Construction Project; MGERP: Mining Geo-Environmental Restoration Project; ECCP: Eco-corridor Construction Project; DP: Desertification Project; CSECP: Comprehensive Soil Erosion Control Project; TP: Terracing Project; HSFCP: High-standard Farmland Construction Project; BTSSP: Beijing–Tianjin Sandstorm Source Project; NSTP: Naked Slope Treatment Project.

### 2.2. Data Sources

Topographic, digital elevation model (DEM), land use, meteorological, soil, and ecological restoration project data are among the data used in this study. The land use data comprise the CAS LUCC land use/cover data, including six primary land use types: cropland, forest land, grassland, watershed, urban and rural residential land, and unutilized land. The Resource and Environmental Science Data Centre (RESDC) provided rainfall and DEM data, the National Earth System Science Data Centre (NESDC) provided meteorological data, HWSD provided soil data, and the National Earth System Science Data Centre (NESDC) and Fugu County Department of Natural Resources and Planning provided ecological restoration project data. All data were first converted in ArcGIS 10.2 to the raster data projection coordinate system, spatial resolution, and data format to make them consistent, then data projection was unified as WGS_1984_Ablers, and the data needed to evaluate the three ESs were unified to 30 m raster data spatial resolution using ArcGIS's resampling tool.

### 2.3. Methodology

### 2.3.1. Evaluation of Ecosystem Services

This study evaluated three typical ecosystem services (ESs) in Fugu County, which face significant ecological challenges such as high erosion risk, water scarcity, and low vegetation cover. The evaluation focused on CS, WC, and SC, and the procedure for calculating each ES is detailed in Table 1. When quantifying ecosystem services using the InVest model, it is first necessary to transform and unify the projected coordinate system, spatial resolution, and data format of the raster data, with the data projection united as WGS_1984_Ablers, and then unify the spatial resolution of the raster data through the ArcGIS resampling function. When applying the model, it is necessary to import data such as DEM (m), average annual precipitation (mm), average annual reference evapotranspiration (mm), depth of the soil limiting layer (mm), effective moisture content of vegetation, land use type, data of subwatersheds, table of biophysical attributes, etc., into the corresponding module of the InVEST model after trimming and other processing. The carbon storage service, soil conservation service, and water yield service were calculated using the model, for which the amount of water conservation needed to be corrected by the water balance equation based on the water yield.

**Table 1.** Approaches and calculation procedure for the estimation of ecological services.

| ESs | Method | Calculation Process |
|---|---|---|
| Water conservation (WC) | The research utilized the water production component of the InVEST model to compute the water yield, which was calculated according to the discrepancy between annual evapotranspiration and actual precipitation [30]. Then, the values were corrected. | The following is the WY calculation formula: $WY_x = (1 - \frac{AET_x}{P_x}) \times P_x$ $WC = \min\left(1, \frac{249}{Velocity}\right) \times \min\left(1, \frac{0.9 \times TI}{3}\right)$ *WY*: water yield per year; *AET*: yearly actual evapotranspiration; *P*: yearly precipitation; WC: Water conservation per year; Velocity: flow rate coefficient; TI: topographic index; Ksat: Saturated soil hydraulic conductivity. |
| Soil conservation (SC) | The study computed both the present and potential soil erosion using the Sediment Delivery Ratio module in the InVEST model based on RUSLE to calculate SC [31]. | The following is the SC calculation formula: $SC = PKLS - USLE = R \times K \times LS \times (1 - C \times P)$ *SC*: the soil conservation per unit area; *PKLS*: the quantity of possible soil erosion per unit area; *USLE*: the actual quantity of soil erosion per unit area; *R*: the erosivity factor caused by rainfall; *K*: the issue of soil erodibility; *LS*: the factor of the slope length; *C*: the management parameters and crop/vegetation cover; *P*: the factor of erosion control practices. |
| Carbon storage (CS) | The carbon storage was estimated by combining the carbon pools of aboveground biomass, belowground biomass, soil organic matter, and dead organic matter using the carbon module of the InVEST model in this work [30]. | The following is the CS calculation formula: $CS = C_{above} + C_{below} + C_{soil} + C_{dead}$ *CS*: the entire amount of stored carbon; $C_{above}$: the carbon density of biomass found aboveground; $C_{below}$: the carbon density of biomass found belowground; $C_{soil}$: the carbon density of organic matter in the soil; $C_{dead}$: the storage of carbon in decomposing organic materials. |

ESs: ecosystem services; CS: carbon storage; SC: soil conservation; WC: water conservation.

### 2.3.2. Quantification of Trade-Offs and Synergies among Ecosystem Services

This study utilized the difference comparison method to evaluate the dynamic trade-offs and synergies between various ecosystem services. The difference comparison method determines whether the relationship between ESs is a trade-off or synergy by comparing the direction of change of two ESs in different years separately. The trade-off synergistic relationship obtained using this method is a binary variable that can reflect spatial hetero-

geneity. This involved comparing the changes in two ecosystem services in 1990, 2000, 2010, and 2020. A positive product of the changes indicated synergies, while a negative product indicated trade-offs. Equation (1) represents the change in the first ecosystem service from period T1 to period T2. Equation (2) represents the change in the second ecosystem service from period T1 to period T2. Equation (3) represents a synergistic relationship between two ecosystem services if the change in the value of both ecosystem services is positive. Equation (4) represents a trade-off relationship between two ecosystem services if the change in the value of both ecosystem services is negative.

$$E_{T1} - E_{T2} = \Delta E \tag{1}$$

$$F_{T1} - F_{T2} = \Delta F \tag{2}$$

$$\Delta E \times \Delta F \geq 0 (synergy) \tag{3}$$

$$\Delta E \times \Delta \leq 0 (trade-off) \tag{4}$$

where *E* and *F* represent the two ecosystem services' respective values; *T*1 and *T*2 represent two different periods; $\Delta E$ and $\Delta F$ represent the alterations in the values of the two ecosystem services during the period from *T*1 to *T*2.

2.3.3. Analysis of the Correlation between ESRs and Ecological Restoration Project Factors

Boosted regression tree modeling (BRT) can be viewed as an additive regression model compared to traditional statistical models. It combines the advantages of regression trees and augmentation algorithms, which can effectively remove interactions between independent variables, fit complex nonlinear relationships, and better explain and predict the interrelationships between variables that are nonlinear, which can help in identifying crucial ecological restoration project elements that have a higher influence on ESRs. In recent years, numerous studies in the domains of geography, ecology, economics, and environmental sciences have made extensive use of BRT because of its usefulness for studying the interactions between complex factors. This study examined the nonlinear link between ecological restoration project factors and ESRs using the BRT model. The augmented regression tree model was then utilized for mechanism exploration. Eleven ecological engineering factors including the Grain for Green Project (GGP), Check Dam Project (CDP), High-Standard Farmland Construction Project (HSFCP), Terracing Project (TP), Sloping Cropland Ecological Construction Project (SCECP), Mining Geo-Environmental Restoration Project (MGERP), Eco-corridor Construction Project (ECCP), Desertification Project (DP), Comprehensive Soil Erosion Control Project (CSECP), Beijing–Tianjin Sandstorm Source Project (BTSSP), and Naked Slope Treatment Project (NSTP) were used as independent variables, and two pairs of synergistic trade-off relationships, CS–SC and CS–WC, respectively, were used as dependent variables.

In this study, eleven ecological restoration projects during 1990–2020 were first organized into panel data using subwatersheds as units. Before applying the BRT model, the covariance test between independent variables was carried out using the Variance Inflation Factor (VIF) test based on SPSS 25, and all independent variables could be applied to the model. The model was then computed using the "gmb" and "dismo" packages in R 4.0.5. The bag fraction (BF), tree complexity (TC), and learning rate (LR) are three crucial BRT model parameters. For this research, the ultimate optimal values for BF, TC, and LR were established at 0.005, 0.5, and 0.005, respectively. Furthermore, half of the data were utilized for training, and the best model was chosen using 10-fold cross-validation.

## 3. Results

### 3.1. Spatial and Temporal Distributions of Ecosystem Services

As shown in Figure 2, the overall spatial trend of WC in Fugu County is decreasing from southwest to northeast. As seen in Figure 3, the regions with high value are mostly

distributed in the Shima River basin, the Gushan River basin, and part of the Qingshui River basin, while the areas with low value are mainly in the Huangfu River basin, and the Shima River's water content measures more than 170 mm. As shown in Figure 4, the value of WC in Fugu County from 1990 to 2020 shows a pattern of decline followed by an increase, with the lowest value of WC in 2000 at 22,652.92 mm and the highest value of WC in 2020 at 82,974.07 mm, which is an increase of 26.28% year on year.The WC in 2000 is lower overall, ranging from 50 to 80 mm, and the WC values in the Gushan and Shima River basins are greater than those in the Beiniu River basin. Overall, WC in 2000 was lower, ranging from 50 to 80 mm, with higher values seen in the Gushan and Shima River basins than in the Huangfu and Beiniu River basins. In 2010, the regions with high WC were limited to the western areas of the Shima River basin, the Gushan River basin, and a section of the Qingshui River basin, receiving 110–170 mm of water. Conversely, the areas with low WC were primarily found in the Huangfu River basin.

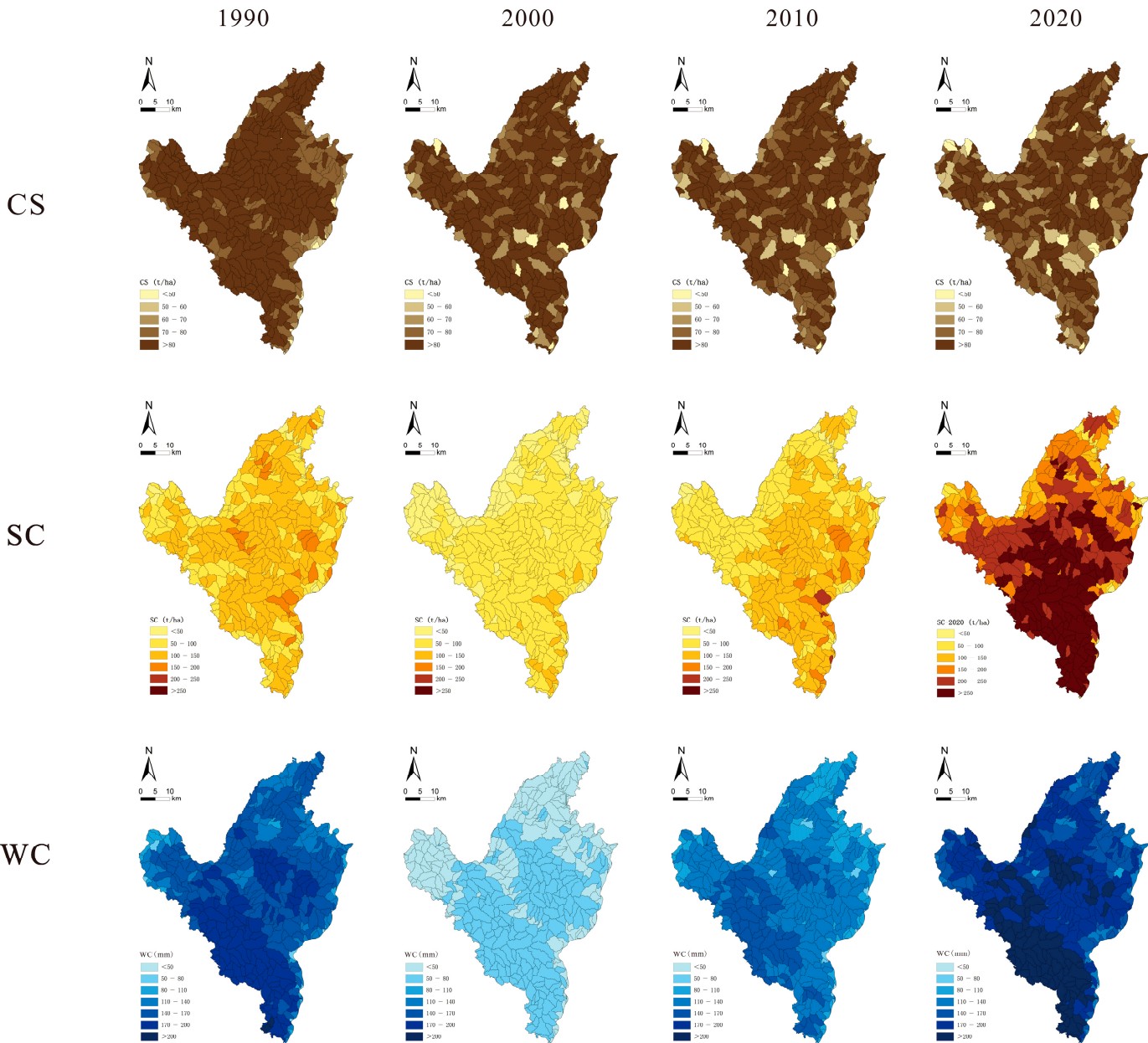

**Figure 2.** Spatial distribution of ecosystem services in Fugu County.

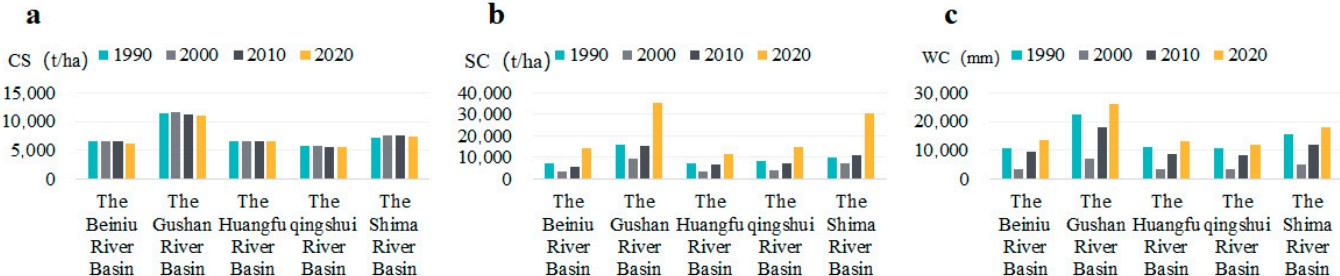

**Figure 3.** Changes in total ecosystem services in five major basins of Fugu County in 1990, 2000, 2010, and 2020. (**a**) Changes in carbon storage services in five major watersheds in Fugu County in 1990, 2000, 2010, and 2020. (**b**) Changes in soil conservation services in five watersheds in False Valley County 1990, 2000, 2010, and 2020. (**c**) Changes in water conservation services in five major watersheds in Fugu County in 1990, 2000, 2010, and 2020.

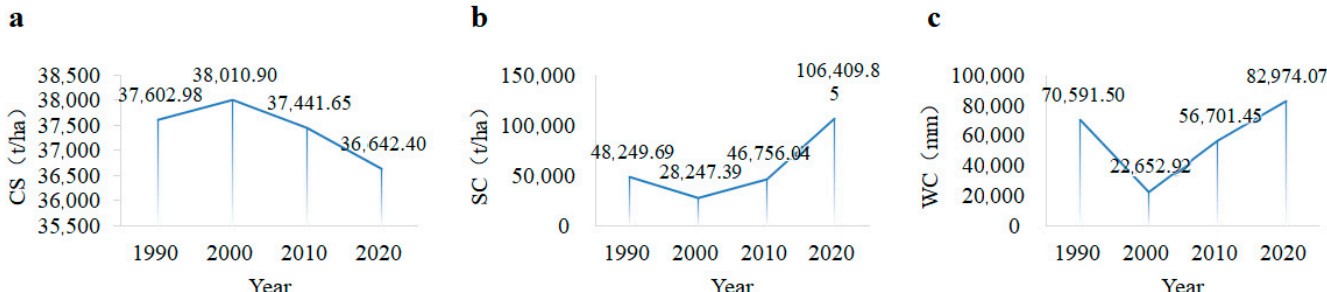

**Figure 4.** Changes in ESs in Fugu County in 1990, 2000, 2010, and 2020. (**a**) Changes in carbon storage services in Fugu County in 1990, 2000, 2010, and 2020. (**b**) Changes in soil conservation services in Fugu County in 1990, 2000, 2010, and 2020. (**c**) Changes in water conservation services in Fugu County in 1990, 2000, 2010, and 2020.

As shown in Figure 3, the value of SC from 1990 to 2020 shows a decreasing and then increasing trend, reaching a minimum value of 28,247.39 t/ha in 2000 and then increasing to a maximum value of 106,409.85 t/ha in 2020. As shown in Figure 2, while the value of SC in 2020 demonstrates an overall decline moving northward from the south, with a notable increase in 2020 compared to 1990, 2000, and 2010, the value of SC in 2000 and 2010 shows a declining trend from the southeast to the northwest. In 2020, there is a decline in the trend from the southern region to the northern region, and the SC value has notably risen in 2020 compared to 1990, 2000, and 2010. The Gushan River basin and the Shima River asin are the only regions with high SC values, as seen in Figure 3.

As shown in Figure 4, the value of CS was stable from 1990 to 2020, with weak fluctuations, first showing a small increasing and then decreasing trend. The value of CS was the highest in 2000, reaching 38,010.90 t/ha, and then began to gradually decline, and it dropped to the lowest in 2020, at 36,642.40 t/ha, a year-on-year decrease of 3.60%, uniformly distributed throughout the whole area of Fugu County. As shown in Figures 2 and 3, in 1990, the Shima River basin, Qingshui River basin, Beiniu River basin, and Gushan River asin as a whole constituted a region of high values, and the values of CS in most of the subwatersheds were above 80 t/ha, with the Huangfu River basin having a lower CS in comparison with the other basins. The CS capacities of the five major basins in 2000, 2010, and 2020 were relatively closed to each other.

### 3.2. Spatial and Temporal Trends in the Trade-Offs and Synergies among Ecosystem Services

In analyzing the trade-offs and synergies between ecosystem services, three sets were identified by comparing the number of subwatershed units exhibiting these phenomena. These can be categorized into two types as depicted in Figure 5: (I) synergistic dominance, such as water conservation (WC)–soil conservation (SC), and (II) trade-off dominance, such

as soil conservation (SC)–carbon storage (CS) and water conservation (WC)–carbon storage (CS). The synergies between SC and WC are attributable to shared influencing factors, including the climate and soil texture.

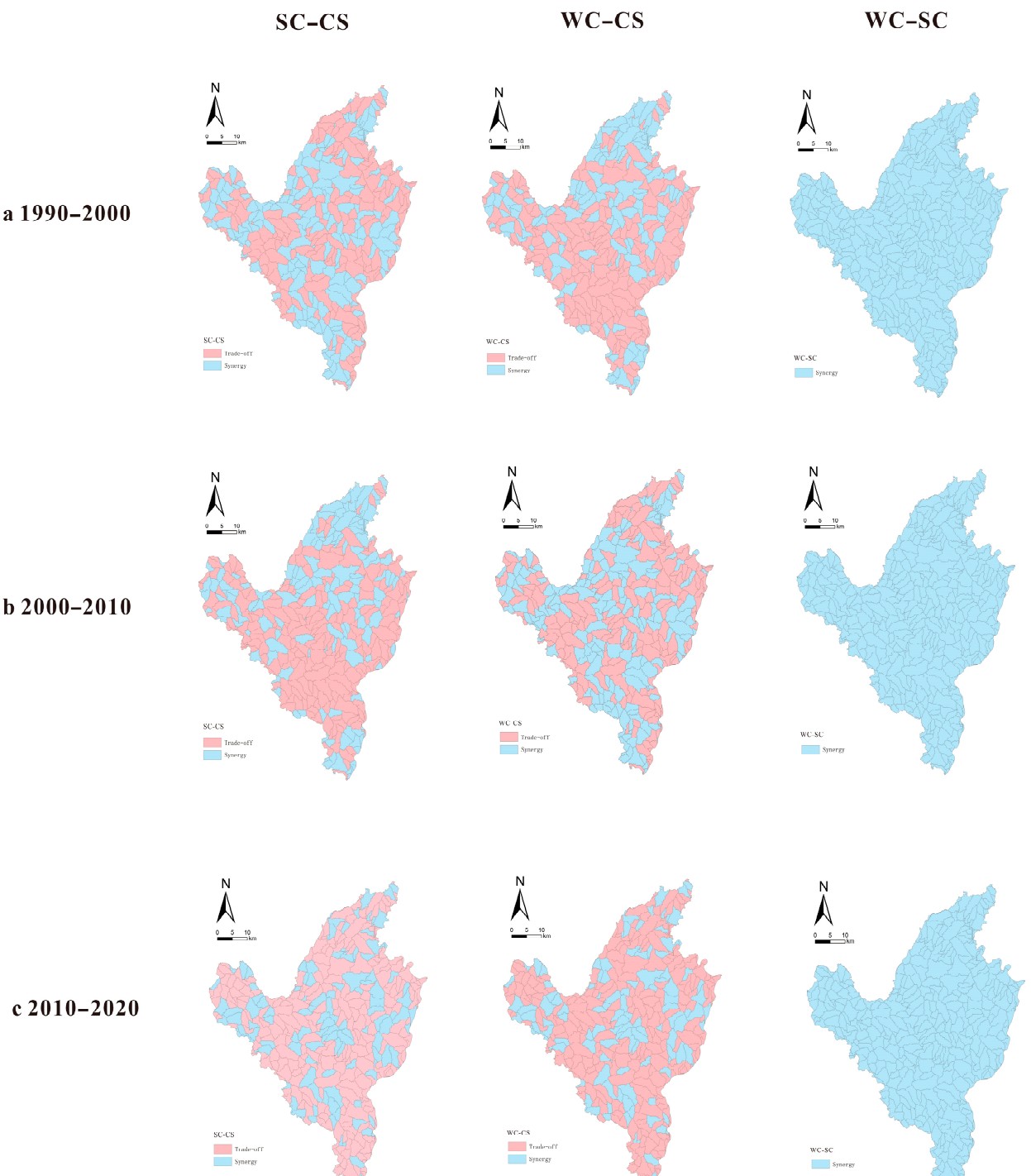

**Figure 5.** Spatial patterns of ESRs in (**a**–**c**) 1990–2020.

The primary spatial trade-offs of CS–SC and CS–WC exhibit strikingly similar spatial distributions. This is primarily because regions with robust crop production (CS) are typically forested areas with high soil conservation (SC) and minimal risk of soil erosion. However, these sites experience high evapotranspiration, which leads to a low water yield (WC), resulting in trade-offs between CS–SC and CS–WC.

Between 1990 and 2020, the trade-offs between SC–CS and WC–CS gradually increased from the Huangfu River watershed in the northern part of Fugu County to the Shima River basin in the southern part of the same county. The trade-offs between SC–CS and WC–CS were most pronounced in the southern part of the Shima River during 2000–2010. The southwestern region is heavily forested, and these woodlands contribute to reducing surface runoff. The majority of the county's agricultural land is located in the central and northeastern parts of the county, which can enhance soil conservation (SC) due to the reduction in erosion in this area. However, the use of water for agriculture may lead to a low water yield (WC), thus creating a trade-off.

### 3.3. Impact of Ecological Engineering Factors on Synergies and Trade-Offs between ESs

3.3.1. The Contribution of Each Ecological Restoration Project Factor to ESRs

The contribution of each ecological restoration project factor to ESRs obtained through the BRT model is shown in Figure 6; the results show that GGP, CDP, HSFCP, and TP are the four most important factors affecting ESRs. At the same time, DP, MGERP, and BTSSP have a low relative relevance. GGP, CDP, HSFCP, and TP account for more than 75% of the contribution of ESRs, leaving less than 25% for ecological restoration project factors. For CS–WC, GGP has the largest independent contribution (35.01%), followed by CDP (22.28%), HSFCP (19.18%), TP (12.29%), and NSTP (3.95%). For CS–SC, GGP had the highest independent contribution to the CS–SC trade-off (34.58%), followed by CDP (22.66%), HSFCP (18.95%), TP (12.43%), and NSTP (4.14%). For both CS–WC and CS–SC, the importance rankings of the ecological restoration project factors of GGP, CDP, HSFCP, TP, NSTP, CSECP, SCECP, and ECCP were the same, while the relative impact of BTSSP on CS–WC was greater than that of its relative impact on CS–SC. The relative impacts of NSTP and CSECP on both CS–WC and CS–SC were all similar to each other. Thus, GGP, CDP, HSFCP, and TP are the four most important ecological restoration project factors that significantly affect ESRs.

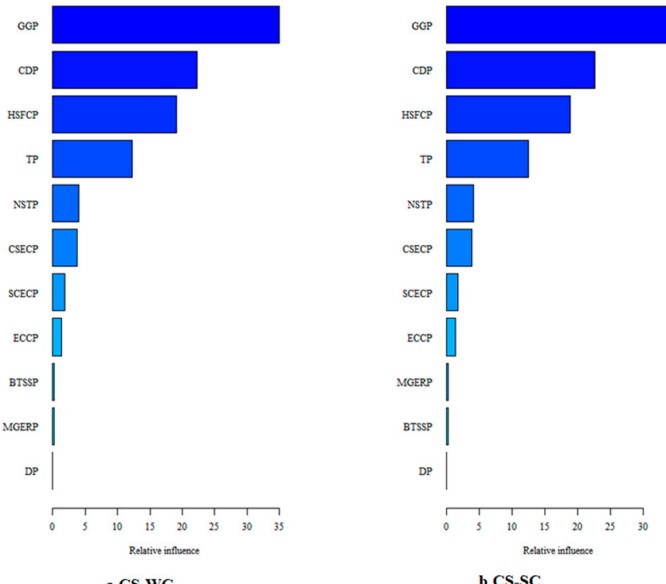

**Figure 6.** The relative contribution of each ecological restoration project. GGP: the Grain for Green project; CDP: Check Dam Project; SCECP: Sloping Cropland Ecological Construction Project; MGERP: Mining Geo-Environmental Restoration Project; ECCP: Eco-corridor Construction Project; DP: Desertification Project; CSECP: Comprehensive Soil Erosion Control Project; TP: Terracing Project; HSFCP: High-standard Farmland Construction Project; BTSSP: Beijing–Tianjin Sandstorm Source Project; NSTP: Naked Slope Treatment Project. (**a**) Ranking of the contribution of each ecological restoration project to the relationship between carbon storage and water conversation; (**b**) Ranking of the contribution of each ecological restoration project to the relationship between carbon storage and soil conservation.

### 3.3.2. Nonlinear Relationship between ESRs and Significant Ecological Restoration Projects

The analysis of the bias dependency plot based on the BRT model shows that the relationship between important ecological restoration project factors and ESRs in Fugu County is characterized by significant nonlinearity, as shown in Figure 7, and the trend of different factors varies. The GGP, CDP, HSFCP, and TP show fluctuating downward, fluctuating downward, fluctuating upward, and fluctuating downward patterns, respectively.

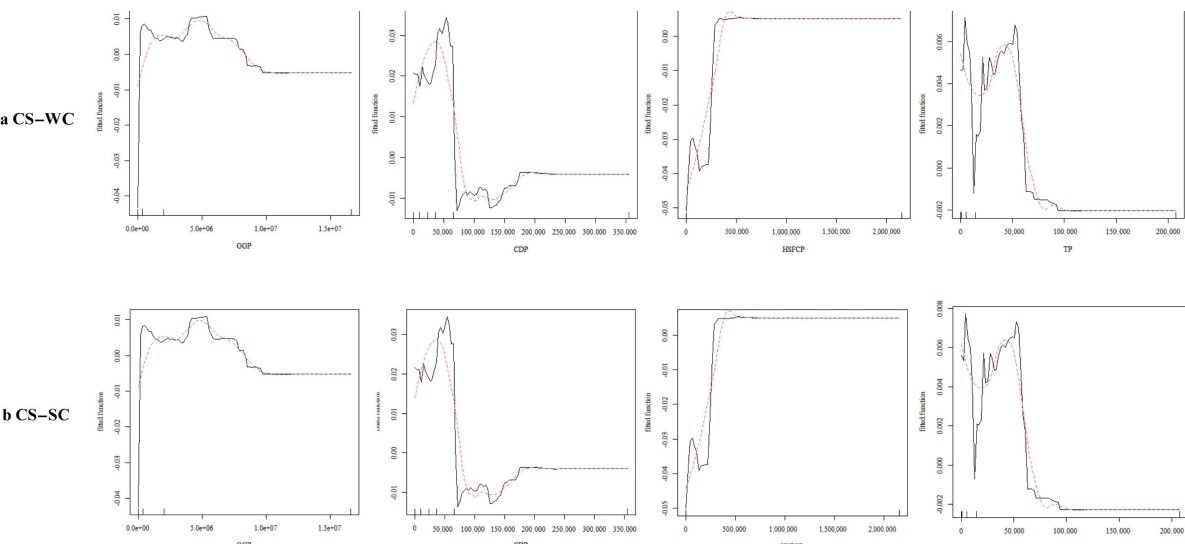

**Figure 7.** Relationship between each ecological restoration project and ESRs. (**a**) Non-linear effects of ecological restoration projects on the relationship between carbon storage and water conservation; (**b**) Non-linear effects of ecological restoration projects on the relationship between carbon storage and soil conservation. The solid line in the figure represents the actual non-linear process of change in the impact of ecological restoration projects on the trade-off synergistic relationship between ecosystem services, and the dashed line represents the fitted non-linear process of change.

The impact of GGP on CS–WC and CS–SC shows a fluctuating downward pattern, with a smoother overall fluctuation of the curve, and the overall impact on ESRs is a decrease in trade-offs and an increase in synergies. As the amount of project implementation increases, GGP will first increase the trade-offs by a small amount, and then the trade-offs will be reduced and the synergies will be increased by a larger amount, likely due to the drastic increase in water use caused by a large increase in the amount of vegetation at the early stage of GGP project implementation, which brings about the problem of balancing trade-offs between various ecosystem services. The impact of CDP on CS–WC and CS–SC shows a fluctuating downward pattern, with more dramatic overall fluctuations in the curves, and the overall impact on ESRs is a decrease in trade-offs and an increase in synergies. At the beginning of the project implementation, CDP will show an increase in trade-offs and then a more substantial decrease in trade-offs, and as the project continues to increase, the impact of CDP on ESRs gradually smooths out after small fluctuations. The impact of HSFCP on CS–WY and CS–SC shows a fluctuating upward pattern, the implementation of HSFCP reduces the synergy between ESs, and then the impact on ESRs becomes smooth, and the overall impact on ESRs is a reduction in synergy and exacerbation of trade-offs. The impact of TP on CS–WY and CS–SC shows a fluctuating downward pattern, the overall fluctuation of the curve is more violent, and the overall impact on ESRs is a reduction in the trade-offs and increase in synergy. The impact of TP on ESRs first shows small fluctuations, and then the curve shows a violent decline, and the impact of TP on CS–WY and CS–SC is more obvious. The impact of TP on ESRs first shows a small fluctuation; the curve then shows a sharp decline, and the impact of TP on CS–WY and CS–SC is more obvious, which significantly reduces the trade-off between ESs, and then the curve gradually smooths out.

## 4. Discussion

### 4.1. Ecological Restoration Project Impacts Trade-Offs and Synergies between the Ecosystem Services Pathways

GGP has planted forests and grasses on cultivated land by providing cash and food subsidies to increase vegetation cover, control soil erosion, and improve farmers' livelihoods. This project has played an important role in increasing vegetation cover, controlling soil erosion, and promoting nonfarm employment. However, it has also been found that this project may adversely affect other ecosystem services, leading, for instance, to a reduction in the area of cultivated land, threatening food security, exacerbating social inequality, and possibly even accelerating water scarcity and affecting the water–sand relationship in the Loess Plateau [32]. The results of this paper demonstrate the impact of GGP on ecosystem services as well as ESRs, which is in line with other studies. Secondly, there is a multiscale impact of ESRs on the ecological benefits of major ecological projects, which provides new ideas and methods for systematically assessing the ecological benefits of major ecological projects. For example, during the implementation of the Three North Protective Forests and the Beijing–Tianjin Wind and Sand Source Control Project in the mountainous areas of the Haihe River basin, there is a prominent trade-off relationship between wind and sand control and water supply services. With the extension of the ecological restoration time, the trade-off relationship leads to multiple spatial scale effects [33]. This paper demonstrates that different ecological restoration projects have different effects on ESRs at the small watershed scale.

This study quantified three typical ESs and obtained the spatial and temporal patterns of the three pairs of ESRs. Eleven important ecological restoration projects and ESRs with different nonlinear variations existed between them. The results of this study showed that the water conservation service and the other two services were all trade-off relationships in Fugu County, which is consistent with other studies on the Loess Plateau [28]. This may be due to the fact that the implementation of a large number of ecological restoration projects in the Loess Plateau region has brought about a large increase in vegetation, which has increased water conservation and also consumed a large amount of groundwater. How to balance the trade-off between water conservation services and other services has become a crucial issue to address in the future. Thus, gaining an understanding of the ESRs can aid in guiding the landing of ecological restoration projects.

In addition, the results of the study showed that GGP, CDP, HSFCP, and TP were the four most important factors affecting ESRs, and different ecological restoration projects had different nonlinear change characteristics on ESRs. Since the 1970s, the area has used extensive terracing and check dams [34]. The quantity of ecosystem services is thought to have been significantly impacted by the installation of ecological restoration projects such as check dams, terracing, and the Grain for Grass project [6]. These changes in ecosystem services have also led to changes in ESRs [11]. Engineering works such as silt dams and terraces, and biological management measures such as afforestation, have alternated over the past 60 years in the Loess Plateau of China and are gradually moving towards the development stage of a comprehensive combination of engineering and biological management [34]. In other regions, these engineering works have also been shown to affect ecological processes, which in turn affect changes in ecosystem services and ESRs [14,35].

### 4.1.1. The Path of GGP Affecting the Trade-Offs and Synergies between ESs

GGP is a typical example of ecological compensation, a massive vegetation restoration initiative that the Chinese government has been carrying out since 1999, which is the biggest ecological restoration project in the world in terms of size and funding [36,37]. It has been implemented mainly in the Beiniu River basin and Gushan River basin in Fugu County, with a small number of sporadic distributions in the Huangfu River basin and Qingshui River basin as well. In Lu et al.'s study, the main cause of the rise in carbon stocks in the project area was the execution of ecological restoration programs, and of the six ecological restoration projects, GGP contributed the most to carbon sequestration. This was

primarily because of the project's enormous area and the reduction in soil erosion brought about by extensive afforestation [38]. As previous studies have shown, both afforestation and soil and water measures contribute to carbon sequestration [39]. Zhang et al. (2010) discovered that GGP considerably raised China's soil carbon sink [40]. According to Lu et al. (2018), ecological engineering increased the area of grassland and forest, which increased the region's carbon stock in China by more than 50% [38]. GGP enhances the amount of plant life by building forests that are sustained by water sources, as well as implementing measures for conserving soil and water and restoring vegetation. The latest research has shown that vegetation restoration has different effects on water yield [35], which can increase the retention and infiltration of precipitation and improve the hydrological cycle, which in turn enhances WC. Vegetation restoration can in turn increase the accumulation of soil organic carbon to improve soil fertility [41], increase the stability of the soil structure, and reduce soil erosion, which in turn increases SC. GGP can increase the carbon and nitrogen content of the soil [42], which in turn increases CS. Soil moisture is critical in ecosystems in arid and semiarid regions, and large-scale vegetation restoration can cause a reduction in soil moisture, which in turn leads to a reduction in water yield [43], which in turn involves a trade-off between WC and other services. Studies have shown that increased water use by vegetation accelerates downstream groundwater depletion, which leads to increased water use conflicts and, in turn, increased trade-offs between ESs [29]. In this study, there was a trade-off relationship between water conservation services and two other ecosystem services, which is consistent with other studies [44]. GGP contributed the most to the synergistic effects of trade-offs among all ecological restoration projects, a result that is also consistent with other studies [38]. Consistent with earlier studies, the effect of GGP on ESRs changed from a reduction in trade-offs to an increase in trade-offs [28].

### 4.1.2. The Path of CDP Affecting the Trade-Offs and Synergies between ESs

CDP is an important ecological project for resource utilization and ecological protection and is deployed in almost all subwatersheds in the county. By constructing dams, the check dam project intercepts sediment in the river and prevents sediment from silting downstream; check dams are the most efficient of all hydraulic engineering measures in terms of their capacity to hold floodwaters [45]. The short-lived retention and long-lived stagnation and drainage effects of the CDP will reduce sediment input into the Yellow River while increasing the long flow of the channel, which in turn will increase the WC [46]. Studies have shown that check dams significantly contribute to vegetation carbon sequestration at long time scales. This is because check dams can achieve the effect of indirectly promoting vegetation carbon sequestration by enhancing soil conservation services [14]. The restoration and protection of vegetation in the CDP can reduce runoff and sediment loads [47,48], which, in turn, can increase CS and SC. The contribution of check dams to soil conservation is very remarkable, and they can directly enhance the soil conservation service function; with the increase in siltation years, the promotion of soil conservation becomes stronger and stronger [14]. However, the water storage and regulation function of CDP may also lead to a reduction in downstream water resources [27], which imposes certain limitations on WC and thus generates a CS–WC trade-off. By reducing erosion and improving the soil retention capacity, CDP helps to improve soil conservation services and further promote water conservation services, thus increasing WC–SC synergy.

### 4.1.3. The Path of HSFCP Affecting the Trade-Offs and Synergies between ESs

HSFCP is a farmland improvement project aimed at improving the quality of farmland and the level of sustainable agricultural development, and is implemented primarily in the Huangfu, Qingshui, and Gushan Rivers' central and eastern regions. HSFCP can increase SC by reducing soil erosion and soil erosion through measures such as terrain improvement, building structures for the conservation of soil and water, and improvement of farmland drainage systems [43]. Channel ditch construction and irrigation renovation measures

have the highest interception efficiency among all engineering measures [46]. HSFCP can provide effective water resource utilization and protection, thus increasing WC, while the vegetation restoration and ecological creation measures of HSFCP increase the plant carbon stock in farmland, increasing CS. The implementation of HSFCP in Fugu County is relatively concentrated in time and in a small area, which will lead to a more concentrated loss of water resources during the implementation of the project; thus, there will be a reduction in synergies between WC and other services in the early stage of implementation. At the same time, the ecological effect of the project has a certain lag, and a shift from decreasing synergies to increasing synergies may occur at a later stage.

4.1.4. The Path of TP Affecting the Trade-Offs and Synergies between ESs

TP was a crucial project for conserving soil and water during the 1970s and 1980s. TP can effectively reduce slope erosion through systematic layout alteration and topography change [14]. TP is mainly implemented in the southern Shima River basin of Fugu County. TP adjusts the path of water flow by rationally laying out the roads and drainage systems, thus increasing WC, improving precipitation retention and infiltration, reducing soil erosion, and promoting water utilization and conservation. By transforming the morphology and structure of farmland and constructing a terrace system, the slope and spacing of fields can be increased to reduce the rate of water flow and decrease soil erosion. Vegetation protection and restoration on terraced platforms can improve soil quality and increase SC. Overall, TP can synthesize and increase synergies between ESs, reduce their trade-offs, and contribute to ecosystem stability. However, TP's reasonable drainage and vegetation restoration may limit a certain degree of WC, resulting in a trade-off between different services. Studies have shown that on short time scales of up to 10 years, TP inhibits water production and soil conservation [14], but on longer time scales, TP encourages soil conservation and water production services. Therefore, the inhibitory effect of TP on trade-offs also strengthened with increasing time scales, in line with the findings of our investigation. Abandoned terraces and terraces with steeper slopes may also exacerbate erosion in gully areas [49]. Therefore, it is important to strengthen the postengineering management of TP in gully areas.

*4.2. Limitations*

In this study, other ecological restoration projects with relatively low impacts include various types of treatment, such as the bare slope treatment, mining environment treatment, sandy land treatment, sloping cultivated land treatment, and the treatment of the Beijing–Tianjin wind and sand source. This is because the problem of poor natural conditions, poor soil, inconvenient traffic, and exposed land parcels is common in soil and rocky mountainous areas, loess hills and gullies, mining collapse areas, and exposed slopes along the Yellow River in Fugu County. These areas are difficult to reforest and restore, and have high construction costs. The reality of limited financial investment and difficulty in implementing construction projects has resulted in too few projects being implemented, and the impacts on ESRs are not as significant as other project types. For a more precise analysis of the impacts of these ecological restoration projects, a smaller scale or a control study area could be chosen for a comparative study. Furthermore, in this study, only three key ESs that are closely related to the current situation and ecological problems in Fugu County were selected to start the correlation analysis. In the future, we could further explore whether there is an impact of the trade-offs and synergies between ecological restoration projects and other ecosystem services. This study was conducted using a small watershed as a unit, but the kinds of changes in such relative impacts and nonlinear relationships at other different spatial scales (e.g., other grid sizes, townships, or larger scales, etc.) need to be further explored. In the future, we could consider studying the scale effect of the impact of ecological restoration projects on ESRs and how it affects more comprehensive ecosystem service types, and we could also continue to explore the range of thresholds for the impact of ecological restoration projects on ESRs, which can better guide the implementation of

ecological restoration projects and the formulation of policies. Human needs for ESs cannot be separated from human well-being, and human needs for ESs can be incorporated into relevant research in the future so that research guides ecosystem management to better meet human needs.

## 5. Conclusions

The three main questions raised in the article can be answered in this study, which is instructive for the management of ecological restoration projects:

(1) A total of two pairs of trade-off relationships, CS–WC and CS–SC, were identified, and in Fugu County, SC–WC is mainly dominated by a synergistic relationship. (2) Different ecological restoration projects have different impacts on ESRs, and GGP, CDP, HSFCP, and TP are the four most important factors influencing ESRs. (3) The relationship between the ecological restoration project and ESRs is a nonlinear change, and at different project implementation stages, the effects of ecological restoration projects on ESRs also change nonlinearly.

This study is based on panel data on ecological restoration projects from 1990 to 2020. This study uses boosted regression tree modeling to systematically explore the relationship between ESRs and various ecological restoration projects, which addresses the problem of insufficient research on how ecological restoration projects has affected ESRs in the past. This study provides direct evidence that ecological restoration projects have an impact on ESRs. The methods and ideas of this study can provide some references for other areas where ecological restoration projects are implemented and for studies at different scales. This study contributes to a deeper understanding of the relationship between ecological restoration projects and ESRs, which is conducive to the scientific management of ecological restoration projects.

**Author Contributions:** Conceptualization, Y.J. and R.Z.; Methodology, Y.J. and R.Z.; Software, Y.J.; Formal analysis, Y.J. and Y.L.; Investigation, Y.J., R.Z. and Z.T.; Resources, Y.L., Z.T. and R.H.; Data curation, Y.L., R.Z., Z.T. and R.H.; Writing—original draft, Y.J.; Writing—review & editing, M.X.; Visualization, Y.J.; Supervision, M.X.; Project administration, M.X.; Funding acquisition, M.X. All authors have read and agreed to the published version of the manuscript.

**Funding:** This research was funded by [National Natural Science Foundation of China] grant number [42207530].

**Data Availability Statement:** The original contributions presented in the study are included in the article, further inquiries can be directed to the corresponding author.

**Conflicts of Interest:** The authors declare no conflict of interest.

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
