# Peer review of "How Do Ecological Restoration Projects Affect Trade-Offs and Synergies between Ecosystem Services?"

_land, doi:10.3390/land13030384_

Round 1

Reviewer 1 Report

Comments and Suggestions for Authors

1.    

   Line 12 before., and

2.       line 16 – modelling

3.       line 48 - decades, most of the research

4.       line 97 – Fu et al [21] is missing.

5.       line 172 + 173 – Error! Reference source not found…

6.       line 182 - Error! Reference source not found.

7.       Shifted Figures and texts – Figures 2 and 3, lines 231-235

8.       Spaces between the words and the parenthesis (citation) throughout the article are missing.

Reviewer 2 Report

Comments and Suggestions for Authors

This manuscript investigates an important topic of how ecological restoration projects affect trade-offs and synergies between ecosystem services. The methods are sound and the results are interesting. However, the introduction could be strengthened to better motivate the research questions and highlight the novelty of this study. The literature review seems quite narrow and could be expanded.

- The writing quality needs improvement throughout to enhance clarity and readability. There are issues with grammar, word choice, sentence structure, and flow in places.

- More explanation is needed in the methods section for some steps that are not clear.

- The discussion could be enhanced by providing more interpretation of the results, comparing to previous literature, and highlighting implications.

As mentioned, the literature review and introduction need significant expansion to better motivate the research questions and highlight the novelty of the study. As is, I'm not fully convinced this fills an important gap in the literature.

The methods seem appropriate, but more details are needed in places to enhance reproducibility. And the overall writing quality makes some sections difficult to interpret.

The results provide helpful insights, yet the discussion is quite limited. More in-depth interpretation and comparison to previous studies is expected for publication.

There may be concerns that the study area (Fugu County) is very small/specific. Addressing the implications or generalizability of the results to broader areas could strengthen impact.

Certain aspects of ecosystem services assessment, trade-off quantification, factors affecting ESR, and so on have been investigated in previous studies in China and elsewhere. This study provides an incremental contribution, but may not provide enough advancement for a higher impact journal article.

In summary, I think this is an interesting preliminary study but it would benefit from:

Ÿ  Strengthening the literature review, introduction/motivation, and highlighting the novelty

Ÿ  Enhancing the clarity of explanations in the methodology

Ÿ  Expanding the discussion and linking back to the larger literature

Ÿ  Describing the larger implications beyond just Fugu County

Ÿ  Demonstrating the advancement this provides over previous related studies

Abstract:

- The abstract clearly summarizes the key aspects of the study but has some grammar issues. Suggest revising sentences like "As a depend-ent variable..." to enhance clarity.

- Consider highlighting the key findings in more detail.

Introduction:

- Expand the literature review and provide more background/motivation on the research questions. The introduction currently seems quite narrow.

- Clarify earlier what specific gaps this study aims to address. The significance and novelty of the study could be emphasized more clearly up front.

Methods:

- Provide more details on how the InVEST model was used to calculate ecosystem services.

- Explain the difference-comparison method more clearly. What do Equations 1-4 represent specifically?

- More explanation of BRT modeling may be helpful for readers unfamiliar with this technique. How was it utilized in this study?

Results:

- The results seem sound, but the writing quality makes comprehension difficult in places. Suggest carefully proofreading this section.

- Consider organizing the results into subsections for clarity.

Discussion:

- Expand the discussion of the results and their implications. Compare to previous literature more.

- Discuss limitations of the study and future research needs.

- Please consider to cite this paper, Enhancing soil carbon and nitrogen through grassland conversion from degraded croplands in China: Assessing magnitudes and identifying key drivers of phosphorus reduction

Conclusions:

- Summarize the key conclusions but avoid simply repeating the results.

- Highlight the main implications and significance of the research.

Reviewer 3 Report

Comments and Suggestions for Authors

The manuscript "How do ecological restoration projects affect trade-offs and synergies between ecosystem services?" analyzes the interaction between the ecosystem services relationships correlating them with restoration projects. The main question of the manuscript is whether the ecosystem service relationships vary when an ecosystem is restored. The authors use a large basin in the Loess Plateau region in China to answer that.

In general, the manuscript asks a fascinating question based on the interactions of ecosystem services and how restoration programs modify them. Since many ecosystem services have synergic or adversarial interactions, which have non-linear responses in the region, it is essential to understand how restoration programs modify these interactions.

The analysis of the manuscript using spatially explicit models to evaluate the ecosystem services in the region seems to be very good. Also, the authors used models for comparing ecosystem services over four decades and boost regression tree models to evaluate the interaction with ecological restoration projects.

My main concern is about the writing organization and the use of acronyms. About the organization, the abstract is highly confusing from the beginning. I understand that this is not a simple issue to explain, but for example, to start with the acronym ESR (which I assume means ecosystem service relations but goes after synergy and trade-offs) is complicated for the reader. Also, in the abstract, justifying the research with "has never been directly studied" does not add to the explanation. I suggest the authors modify the abstract, particularly at the beginning, to explain the central issue of the project better.

 The second problem is the among of acronyms that sometimes do not correspond to the concept (such, ESR that was not defined at the beginning of the introduction). The excessive use of acronyms makes it difficult for a reader to understand without checking what they mean. This makes, for example, section 3.3 (lines starting at 261) impossible to read to comprehend the results. 

A third problem is the structure of the introduction. The authors include vast information about methods in the conceptual part of the introduction (lines 52 to 63) and later return to the conceptual discussion. In this section, it also seems there are two goals of the paper: in line 79, "This study investigates..:" and in line 105, "This study aims…" I suggest modifying the structure of the introduction to make it clear.

Figures do not have enough information. For example, in Figures 3 and 4, I could not find what a, b, and c meant in any of the areas.

Finally, it needs a format revision. There are several areas where a sentence starts after a comma (for example, lines 319 and 320) or at the end of line 172, which is marked with an error.

Comments on the Quality of English Language

I included my comments in the section above.

Round 2

Reviewer 2 Report

Comments and Suggestions for Authors

The manuscript has been well improved and in a good shape now.

Comments:
1. Author's affiliations should be corrected.
2. Please ensure use high quality figures during the publication.
3. L251, Error! Reference source not found.
